# Pooled genome-wide CRISPR screening for basal and context-specific fitness gene essentiality in *Drosophila* cells

Raghuvir Viswanatha[1]*, Zhongchi Li[1,2], Yanhui Hu[1], Norbert Perrimon[1,3]*

[1]Department of Genetics, Harvard Medical School, Boston, United States; [2]School of Pharmaceutical Sciences, Tsinghua University, Beijing, China; [3]Howard Hughes Medical Institute, Boston, United States

**Abstract** Genome-wide screens in *Drosophila* cells have offered numerous insights into gene function, yet a major limitation has been the inability to stably deliver large multiplexed DNA libraries to cultured cells allowing barcoded pooled screens. Here, we developed a site-specific integration strategy for library delivery and performed a genome-wide CRISPR knockout screen in *Drosophila* S2R+ cells. Under basal growth conditions, 1235 genes were essential for cell fitness at a false-discovery rate of 5%, representing the highest-resolution fitness gene set yet assembled for *Drosophila*, including 407 genes which likely duplicated along the vertebrate lineage and whose orthologs were underrepresented in human CRISPR screens. We additionally performed context-specific fitness screens for resistance to or synergy with trametinib, a Ras/ERK/ETS inhibitor, or rapamycin, an mTOR inhibitor, and identified key regulators of each pathway. The results present a novel, scalable, and versatile platform for functional genomic screens in invertebrate cells.
DOI: https://doi.org/10.7554/eLife.36333.001

**\*For correspondence:**
ram@genetics.med.harvard.edu (RV);
perrimon@receptor.med.harvard.edu (NP)

**Competing interests:** The authors declare that no competing interests exist.

## Introduction

Systematic perturbation of gene function in eukaryotic cells using arrayed (well-by-well) reagents is a powerful technique that has been used to successfully assay many fundamental biological questions such as proliferation, protein secretion, morphology, organelle maintenance, viral entry, synthetic lethality, and other topics (*Mohr et al., 2014*). An alternative approach, widely used in mammalian cells, is pooled screening that uses limited titers of integrating lentiviral vectors carrying a perturbative DNA sequence such that each cell receives one integrating virus. In pooled screens, the perturbing DNA reagent serves as the tag in subsequent sequencing (*Berns et al., 2004*; *Moffat et al., 2006*; *Brummelkamp and Bernards, 2003*). A key benefit of this approach is that pool size can be extremely large, allowing high reagent multiplicity and thus increased screen quality. The pooled approach in mammalian cells has been used extensively to perform RNAi and more recently single guide RNA (sgRNA) screens using CRISPR/Cas9 (*Shalem et al., 2014*; *Wang et al., 2015*; *Hart et al., 2015*).

Genetic loss-of-function arrayed RNAi screens in *Drosophila* cell lines have provided insight into genes regulating various biological processes (*Boutros et al., 2004*; *Björklund et al., 2006*; *Kiger et al., 2003*; *D'Ambrosio and Vale, 2010*; *Bard et al., 2006*; *Guo et al., 2008*; *Hao et al., 2008*; *Housden et al., 2015*). However, this approach has drawbacks that limit resolution, including off-target effects and incomplete loss-of-function due to RNAi, and the high cost of reagent multiplicity and replication due to the arrayed format. Pooled CRISPR may address the major drawbacks: CRISPR generates complete loss-of-function alleles and causes fewer off-target effects on average (*Morgens et al., 2016*; *Evers et al., 2016*), and the pooled format allows greater multiplicity and replication for unit cost. Approximately half of the genes in *Drosophila*, arguably the best

**eLife digest** Genes are made up of DNA and carry the instructions necessary to build an organism. Humans have over 20,000 genes, while other animals, such as fruit flies, have about 14,000. An ongoing challenge in biology is to identify the role of every gene in the human body. Since most of them are conserved in the fruit fly, this insect is one of the most extensively studied organisms.

Scientists often use a technique called CRISPR to edit genes. It enables researchers to modify DNA sequences to selectively alter the purpose of a gene or even turn it off to find out what it does. CRISPR requires a guide molecule (for example, sgRNAs), which leads the system to a particular DNA sequence to start the process. Often, researchers create many sgRNAs and deliver them to a large pool of cells with the help of viruses, so that each cell gets a different sgRNA that mutates a different gene. When the cells are then treated with a specific drug, the composition of the sgRNAs in the pool changes, depending on which genes are needed to withstand the drug, and which genes – when turned-off – create cells that are resistant to the drug.

Although thousands of mutant flies have been created to investigate how a deactivated or faulty gene can affect the health and behavior of the fly, we still lack meaningful information on about half of their genes. This is partly because the viruses used to deliver sgRNAs in mammals do not work in fly cells. Here, Viswanatha et al. developed a simple protocol to generate cell pools of CRISPR mutants, which uses a new strategy that uses bacteria to deliver DNA to fly cells.

This allowed to identify over 1,000 genes necessary for cells to multiply properly, many of which had not been studied before. The technique was also used in combination with drugs to examine the interactions between genes and drugs – an approach that could be further adapted to examine interactions between genes and nutrients, or between genes.

This new approach will open doors to systematically uncover the purpose of every gene in the fly. A better understanding of what genes do could help to identify potential genetic weaknesses in certain types of cancer or other diseases, which may lead to the development of more effective treatments. Moreover, the method is likely to work in other insects, for example, mosquitos, where it may uncover new genes involved in mosquito-borne diseases such as malaria or Zika virus.
DOI: https://doi.org/10.7554/eLife.36333.002

characterized multicellular genetic model system, lack functional characterization (*Ewen-Campen et al., 2017*), so the need to develop orthogonal screening approaches is clear.

By comparing the abundance of guides present in actively growing populations of cells at different time points during growth, CRISPR screens provide a relative measurement of cell doubling, but because the cause of proliferation reduction is unclear from the screen alone, the screens are said to identify genes necessary for optimal fitness rather than essential genes (*Hart et al., 2015*). Essential genes, those absolutely required for cell doubling, are, by definition, a subset of fitness genes.

Here, we introduce a new method to deliver pooled DNA libraries stably into cell lines. We use this technology to conduct a genome-wide CRISPR screen for optimal fitness in *Drosophila* cells and identify 1235 genes essential for fitness, 303 of which are uncharacterized in *Drosophila*. Moreover, we show that the system can be used in combination with drug perturbation to identify genes that when knocked out buffer cells against the drug or act synergistically with it. The method should be amenable to adapting any pooled DNA library screening approach to *Drosophila* or other invertebrate cell lines, such as shRNA knockdown (*Berns et al., 2004*) or CRISPR activation/inhibition.

## Results

### Development of a pooled library delivery method for *Drosophila* cell-lines

Pooled mammalian cell-line screens use lentiviral vectors to deliver highly complex libraries of DNA reagents. However, the use of lentiviral vectors in insect cells is extremely inefficient (unpublished observations) possibly due to toxicity (*Qin et al., 2010*). An important advantage of library delivery using lentiviral transduction is that each sequence integrates into a transcriptionally active site in the

host chromosome where it is expressed and remains in the genome as a detectable barcode such that enrichment or depletion of each sequence can be monitored by massively parallel (next-generation) sequencing. As an alternative strategy, we assessed the efficiency of phiC31 site-specific recombination mediated by plasmid transfection. Methods for recombination in cell lines were recently developed (*Manivannan and Simcox, 2016*; *Neumüller et al., 2012*; *Cherbas et al., 2015*), but the quantitative efficiency of integration has not been reported. Efficiency is critical for pooled screening applications as it must reach a threshold above which generating and maintaining >1000X representation of a library of tens of thousands of elements becomes technically and cost-feasible (*Kampmann et al., 2014*). We note that a previous study attempted to use pooled transient transfection for barcoded delivery of a DNA library in *Drosophila* cells but failed to identify any essential genes using the system, most likely due to the lack of a mechanism for retention of the DNA reagents in the cells through extended passaging (*Bassett et al., 2015*).

To test phiC31 integration efficiency in *Drosophila* cells, a *Drosophila* S2R + cell line derivative, PT5, harboring a mobilized MiMIC transposon containing *attP* sites flanking mCherry (*Neumüller et al., 2012*) was transfected with a plasmid containing *attB* flanking a GFP-2A-Puro(res) cassette, which we termed 'pLib6.4', along with a phiC31 helper plasmid (*Figure 1A*). The population was then passaged for two months to dilute unintegrated DNA. Importantly, no selection reagent was added during these passages in order to monitor the efficiency of integration rather than the added efficiencies of integration and selection. Integration efficiency, inferred through flow cytometry (*Figure 1B*), suggested that phiC31-mediated cassette exchange occurred in ~20% of the cells (even without accounting for incomplete transfection), 123-fold more than the background illegitimate recombination rate observed without phiC31 (*Figure 1B*). Interestingly, pLib6.4, which contains separated *attB* sites flanking the DNA sequence to be integrated, allowed ~40 fold greater integration efficiency than traditional *Drosophila* attB40 vectors such as pCa4B (*Markstein et al., 2008*), in which the *attB* sites are adjacent and require integration of the entire plasmid (data not shown). Growth in puromycin enriched for integrants (*Figure 1B*).

We next adapted the platform to CRISPR-Cas9 knockout screening. First, we generated PT5 S2R + cells stably expressing metallothionein-driven SpCas9 ('PT5/Cas9'). In combination with an sgRNA targeting the nonessential gene *Dredd*, PT5/Cas9 cells without induction were capable of editing nearly 50% of *Dredd* alleles, which was higher than that achieved by repeated rounds of transient transfection with a SpCas9 expression plasmid (*Figure 1C*) and not improved by copper induction (*Figure 1—figure supplement 1*). In a pilot test of a pooled screening system, we transfected cells with a pool of sgRNAs in pLib6.4 and monitored sgRNA abundance following passaging, reasoning that sgRNAs targeting genes required for optimal fitness would be lost while genes whose presence or absence has no effect on cell growth would be retained (*Figure 1D*). After 60 days (roughly 60 cell doublings), the pools were significantly depleted of some sgRNAs targeting the essential genes *Rho1* and *Diap1* relative to those targeting genes predicted to have non-essential functions (*Figure 1E*). Monitoring *Rho1* or *Diap1*-targeted sgRNAs in the screen pools 30, 45, or 60 days post-transfection showed that 45 days of passaging is optimal (*Figure 1F*). A principle concern of transfection-mediated pooled screening is the potential for low signal-to-noise due to multiple sgRNA delivery to the same cell shortly after transfection, but the subsequent loss of all but one sgRNA at the end of the assay. To determine whether the signal-to-noise ratio could be improved by reducing transfection multiplicity, we developed in parallel an inducible Cas9 expression system in S2R+/PT5 cells using intein-Cas9 (*Davis et al., 2015*) coupled with inducible expression in order to withhold Cas9 activity until sgRNAs have integrated (*Figure 1G*, *Figure 1—figure supplement 2*). Surprisingly, a comparison of dropout efficiencies between the inducible and constitutive Cas9 platforms showed more selective reduction of *Rho1* or *Diap1* sgRNAs with constitutive rather than inducible Cas9, most likely due to the lower overall editing efficiency from intein-Cas9 as previously reported (*Liu et al., 2016*) (*Figure 1G*, *Figure 1—figure supplement 2B,C*). From these results, we conclude that the use of phiC31 integration into the PT5/Cas9 cells is suitable for scalable perturbation screening using a pooled sgRNA library in *Drosophila* cells.

## Genome-wide CRISPR screening in Drosophila cells and screen metrics

To construct a genome-wide sgRNA knockout library for *Drosophila*, we pre-computed sgRNAs for the first half of the coding region of all genes (*Ren et al., 2013*), applied efficiency and frame-shift filters previously shown to correlate with reagent success (*Housden et al., 2015*; *Bae et al., 2014*),

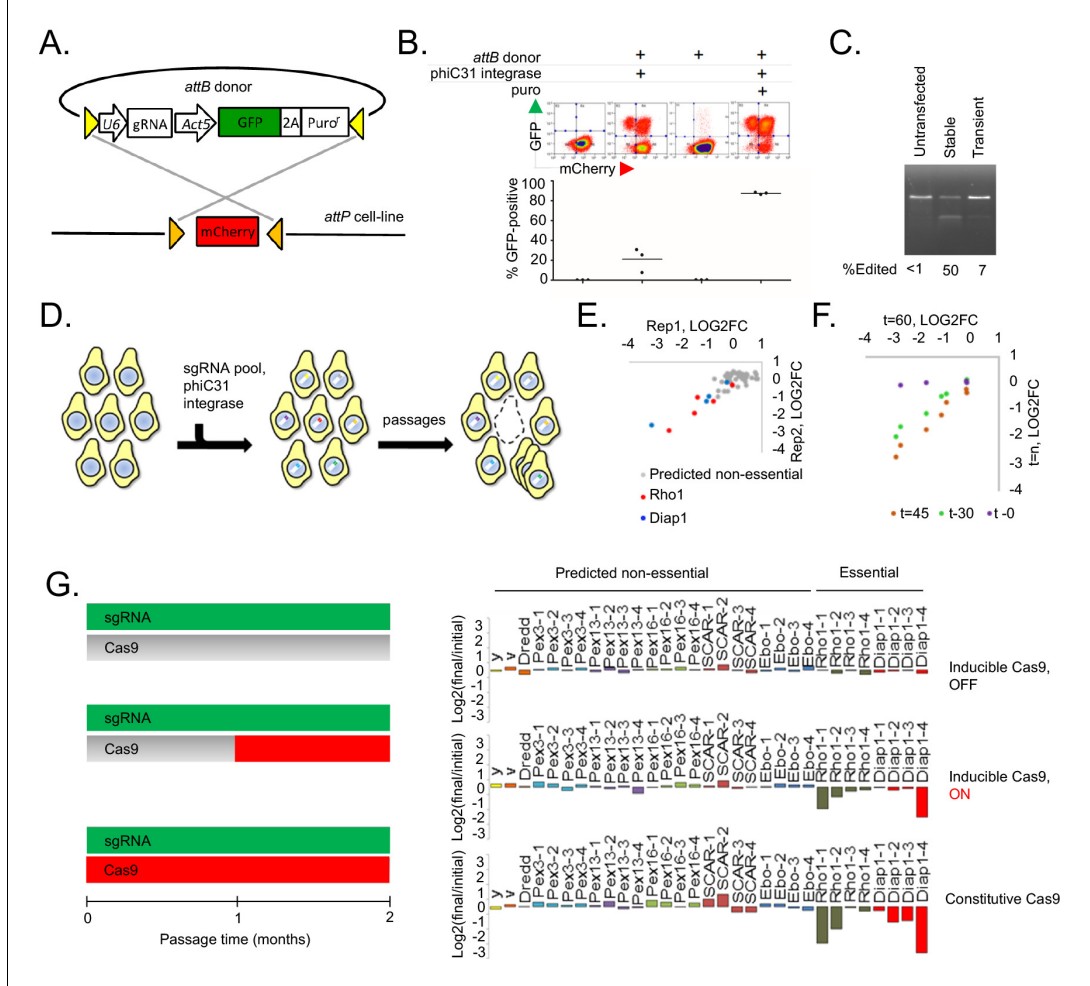

**Figure 1.** A novel method for introducing highly complex DNA libraries using phiC31 recombination. (**A**) phiC31 *attP-attB* recombination strategy. S2R +/PT5 cells containing *attP* sites (gold) flanking mCherry were recombined with *attB* donor (pLib6.4) containing *attB* sites (yellow) flanking U6 promoter for sgRNA expression and GFP-2A-Puro expression cassette. (**B**) Recombination efficiency measured by flow-cytometry. Transfected cells were and grown with or without puromycin as indicated and passaged for 60 days. Graphs reflect total percentage of stable integrants (GFP+/total). N = 3. (**C**) Cells stably or transiently transfected to express Cas9 or control vector were each additionally transfected with an sgRNA targeting the *Dredd* allele followed by editing efficiency assay (T7E1) at the *Dredd* locus. (**D**) Scheme for pooled screens containing a library of integrating sgRNA expression vectors. (**E**) Dropout of essential-gene targeted sgRNAs from a minipool of 31 sgRNAs. Two replicates of PT5 or PT5/Cas9 cells transfected with sgRNAs targeting *Rho1* (red) or *Diap1* (blue) and additional sgRNAs targeting eight genes predicted to have non-essential functions (grey) were passaged with puromycin for 60 days and sgRNA abundance was measured using next-generation sequencing. Graph shows log2(fold-change) of each sgRNA in cells expressing Cas9 divided by sgRNAs in cells not expressing Cas9. (**E**) Optimizing passage time for dropout measurements. sgRNA abundance was detected from cells transfected as in (**D**) but analyzed initially, after 30 days, or after 45 days, and log2 fold-changes were compared to those at 60 days. (**G**) *Left:* Schematic of experiment to test effect of inducible versus constitutive Cas9 activity. *Right:* Dropout efficiencies from pooled screens using inducible versus constitutive Cas9 and a mixture of sgRNAs targeting either essential genes or those predicted to be non-essential. Vertical axis reflects log2(fold-change) for each sgRNA. Shown are means of two independent replicates.

DOI: https://doi.org/10.7554/eLife.36333.003

The following figure supplements are available for figure 1:

**Figure supplement 1.** Copper induction is not required in PT5/MT-Cas9 cells to give maximal gene editing efficiency.
DOI: https://doi.org/10.7554/eLife.36333.004

**Figure supplement 2.** Validation of Cas9 induction system in *Drosophila* S2R + cells.
DOI: https://doi.org/10.7554/eLife.36333.005

**Figure supplement 3.** Design of sgRNA library vector and sgRNA PCR for next-generation sequencing.
DOI: https://doi.org/10.7554/eLife.36333.006

and ranked the remaining designs based on the uniqueness of the seed region sequence (12 bp downstream of the PAM) and the number of potential off-target sites. The top ranked 6–8 sgRNAs per gene (85,558 in total) were chosen and synthesized together on a microarray chip with non-targeting and intergenic negative controls, harvested by PCR, and cloned into pLib6.4 using published methods (*Gilbert et al., 2014*). We used a barcode strategy to separate the full library into three focused library pools so that we could perform focused or genome-wide screens. Each focused subset of the library targets a unique set of experimental genes as well as a common set of controls (*Figure 2A*; *Supplementary file 1*).

To identify fitness genes under basal growth conditions, we transfected cells with pLib6.4 containing the library of sgRNAs along with phiC31 helper plasmid and passaged the cells every 5 days for 45 days. We transfected ~1500 cells per sgRNA and carried >1500 cells per sgRNA per passage to maintain the diversity of the original library. Each experimental library was in-line barcoded using specific primers and sequenced together using next-generation sequencing (*Figure 1—figure supplement 3*). To determine reagent correlation, the three separated sgRNA pools were each transfected and cells passaged independently and common controls were compared (*Figure 2A*; *Supplementary file 1*). Comparing sgRNAs prior to transfection versus 45 days after transfection yielded high reagent correlations between the same sgRNAs as demonstrated for *Rho1* or intergenic guides, for which different sequences produced a highly reproducible range of varied dropout efficiencies (*Figure 2B*; *Supplementary file 1*).

We used MAGeCK (*Li et al., 2014*) maximum likelihood estimation (MLE) to compute fitness Z-scores for each gene from the log2 fold-changes of individual sgRNAs, which yielded good gene-level correlation between sequential biological replicates (*Figure 2C*; Pearson's r = 0.65). Because all true fitness genes must be expressed as mRNA, fitness scores were compared with gene expression from S2R+ cells (*Celniker et al., 2009*) as an orthogonal validation of the CRISPR results (*Figure 2D*). Fitness genes were highly enriched for genes with any expression level of RPKM > 1 (*Figure 2D*). By ranking genes by fitness score, we calculated a rank-wise false-discovery rate (FDR, defined as cumulative distribution of false-positive/[true positive + false positive]) and set a cutoff at 5%, identifying 1235 fitness genes (*Figure 2E*).

The current state-of-the-art for *Drosophila* cell-based screens is arrayed RNAi (*Mohr et al., 2014*). We re-analyzed a genome-wide RNAi viability screen in S2R+ cells (*Boutros et al., 2004*) and removed double-stranded RNA designs with predicted off-targets, which significantly lowered the FDR of the RNAi screen (see Materials and methods). Nevertheless, compared with pooled CRISPR screen results, the re-analyzed RNAi-based fitness gene set still had a far higher FDR, allowing the discovery of only 145 fitness genes at an FDR of 5% or incurring 44% FDR with a fitness gene set of 1235 genes (*Figure 2E*). Among genes with an RPKM > 1, RNAi was more likely to identify highly expressed genes, whereas this bias was less prominent in the CRISPR screen (*Figure 2—figure supplement 1*). We used receiver-operating characteristic (ROC) curves to determine screen sensitivity for conserved, large macromolecular complexes. Whereas CRISPR and RNAi were both able to detect cytoplasmic ribosomal or proteasomal genes, CRISPR was better able than RNAi to identify significant fractions of the mitochondrial ribosome, aminoacyl-tRNA ligases, RNA polymerase II, basal transcription factors, RNA exosome, or the replication fork at low FDR (*Figure 2F*). As a control, neither method detected peroxisome genes, consistent with the observation that cell lines lacking peroxisomes grow normally (*Goldfischer et al., 1973*) (data not shown) (*Figure 2F*). Interestingly, a human CRISPR-RNAi screen comparison also found a similar inability of RNAi to detect genes of the mitochondrial ribosome whereas it exceeded CRISPR sensitivity in detecting genes of the cytoplasmic ribosome, possibly due to differences in mRNA stability (*Hart et al., 2015*).

A technical comparison of CRISPR screens in humans and fly cells indicated that the method performs similarly in both systems. First, the fly screen had similar sensitivity (true-positive rate at 5% FDR) to genome-wide screens in human cell-lines performed with similar reagent number (GeCKO v2 screens) when compared against top RNAi hits (*Sanjana et al., 2014*) (*Figure 2G*). By comparing sgRNAs that dropped out versus those that failed to dropout for known essential genes, we were able to compute a probability-based position matrix for optimal sgRNA design in *Drosophila* (*Figure 2H*). The position matrix shows complementary nucleotide biases at 15/21 positions and is broadly consistent with human CRISPR screens (*Doench et al., 2016*). Specifically, out of the first position of the PAM sequence and the 8 bp seed region before the PAM sequence, 5 of 9 positions

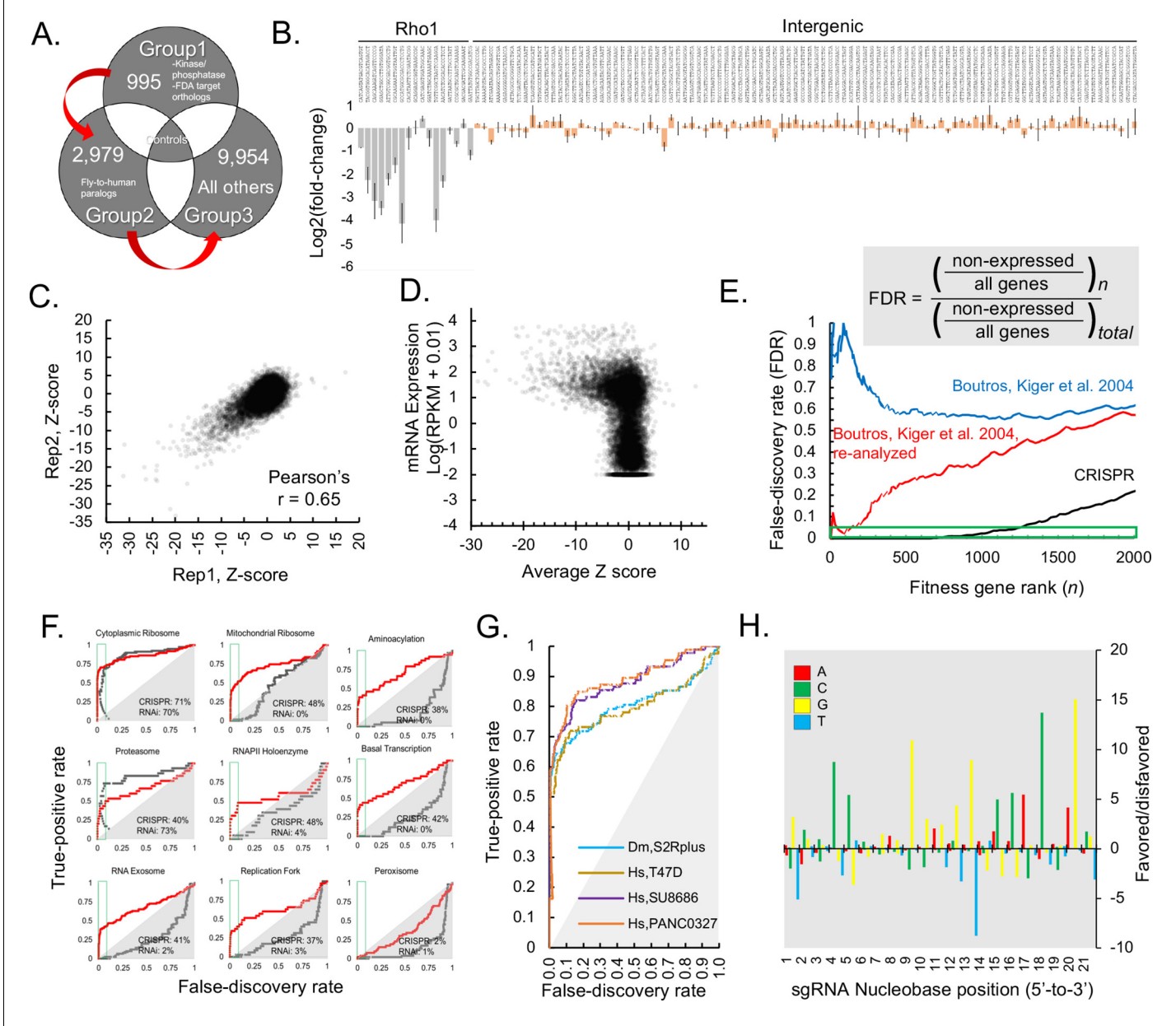

**Figure 2.** Genome-wide CRISPR dropout screen in *Drosophila* S2R+ cells, results and metrics. (A) CRISPR library is maintained in three distinct sublibrary groups as indicated, containing common controls. (B) sgRNA-level analysis of common controls in each group verify similar growth rates during each sublibrary screen. log2(fold-changes) of all sgRNAs representing two common controls, *Rho1* (grey) and intergenic (pink). The average and standard deviation of log2(fold-changes) are shown for all individual sequences corresponding to *Rho1*-targeting positive controls or intergenic negative control sequences. (C) Gene-level analysis of sequential replicate screens. Log2(fold-changes) for all sgRNAs (85,558 in total) were first determined and then aggregated into a single Z-score using the maximum likelihood estimate (MLE) computational approach for each of 13,928 *Drosophila* genes in two independent, sequential replicates and plotted. (D) Z-score was calculated from average of replicate Log2(fold-changes), (*Supplementary file 1*) and these were plotted against RNAseq expression value (log(RPKM + 0.010)) (MODEncode). (E) Rank-wise false-discovery rate (FDR) of pooled CRISPR compared with arrayed RNAi (*Boutros et al., 2004*), original data or following re-analysis (see Materials and methods). Cumulative distribution of false-discovery error at indicated gene rank divided by the total possible false-discovery error, where 'error' is defined as a phenotypic assertion for any gene with RPKM <1. (A) True-positive rate (TPR) for major eukaryotic essential genes shows broader distribution of functional classes revealing fitness essentiality from CRISPR than RNAi screens. Receiver operating characteristic (ROC) curves displaying rate of discovery of components of selected essential eukaryotic complex (*Kanehisa et al., 2017*) as a function of FDR. Curves compare CRISPR knockout screen (this study) with reanalyzed genome-wide RNAi (*Boutros et al., 2004*). (G) True-positive rate (TPR) of *Drosophila* CRISPR screen is in a similar range to TPRs from human CRISPR screens using libraries of similar size. Comparison of true positive rate between human cell-line screens (infected with GeCKO v2) and *Drosophila* CRISPR screening using high-confidence RNAi hits as true positives (*Lenoir et al., 2018*; *Sanjana et al., 2014*;

*Figure 2 continued on next page*

*Figure 2 continued*

*Boutros et al., 2004*). (H) Position matrix for optimal sgRNA design based on CRISPR screen. For the top 500 genes, which were hits with <2% FDR in the CRISPR screen, hypergeometric probability of an A, C, G, or T nucleobase was calculated from strongly depleted ('good', LOG(p-value), above 0 on the y-axis) sgRNA designs versus unchanging ('bad', negative LOG(p-value), below 0 on the y-axis) sgRNA designs.

DOI: https://doi.org/10.7554/eLife.36333.007

The following source data and figure supplement are available for figure 2:

**Source data 1.** CRISPR and RNAi screen comparisons, continued.

DOI: https://doi.org/10.7554/eLife.36333.009

**Figure supplement 1.** CRISPR and RNAi screen comparisons, continued.

DOI: https://doi.org/10.7554/eLife.36333.008

are fully consistent while 2 of 9 position are partially consistent with the Doench score (*Doench et al., 2016*) (*Figure 2H*). The analyses support an underlying mechanistic unity of targeting by Cas9 and NHEJ repair between human and fly cells.

Copy number has been shown to correlate with the CRISPR viability score in some mammalian CRISPR screens, presumably due to induction of greater DNA damage foci for copy-number-amplified (CNA) loci, creating spurious fitness calls (*Meyers et al., 2017*). By contrast, we find that CRISPR in *Drosophila* shows no detectable bias towards CNA genes for 97% of the genome (*Figure 2—figure supplement 1B*). Interestingly, genes with extreme CNA of 8 or more copies (3% of genes) exhibited a bias of ~1.8 fold in the CRISPR screen, but the magnitude of this bias was similar to that seen in an RNAi screen for the same genes (*Figure 2—figure supplement 1B*), suggesting that CNA genes in *Drosophila* cells are enriched for fitness essentiality. Finally, copy-number correction had no detectable effect on FDR, true-positive rate, or enrichment of major macromolecular complexes (not shown). The analysis shows that *Drosophila* CRISPR screens do not have detectable copy-number bias.

## Analysis of *Drosophila* cell fitness genes

We next analyzed fitness genes in *Drosophila* S2R+ cells. We first performed gene ontology enrichment on CRISPR screen hits at 5% FDR. The gene set is enriched for essential processes such as translation, transcription, splicing, etc. and depletion for processes such as chitin metabolism that are necessary in whole flies but not cells (*Figure 3A*). Next, we compared CRISPR hits to fly genes with lethal alleles annotated in FlyBase after removing genes for which no allele has yet been constructed. At 5% FDR,~17% of cell-essential genes overlapped with whole-fly essential genes (*Figure 3B*). This overlap is in a similar range as knockout mouse viability compared with mouse cell-line fitness essentiality (where the overlap is ~27%) (*Bartha et al., 2018*), and may represent different genetic requirements of whole animals versus cell lines or of germline versus somatic cells (*Garcia-Bellido and Robbins, 1983*).

A high-resolution fitness map in fly cells now allows us to compare fitness genes among species. The fly fitness gene set partially but significantly overlaps with characterized fitness gene sets from yeast and human cells (*Figure 3C*). Moreover, an unbiased comparison of gene ontology terms between fly and human fitness genes displays a high degree of correlation (*Figure 3D*; Pearson's r = 0.56). Despite significant overlap between fly and human cell-line fitness genes and conservation of gene ontology terms enriched in both screens, many fly fitness genes map to human genes that were not identified in human CRISPR screens. We hypothesized that fly CRISPR screens identify essential genes in *Drosophila* cells but not in human cells due to paralog redundancy. To test this hypothesis, we inferred putative paralog relationships for moderately or highly conserved orthologs using the DRSC Integrative Ortholog Prediction Tool (DIOPT) (*Hu et al., 2011*) and retrieved fitness scores for the corresponding human gene using genome-wide human cell-line CRISPR data (*Hart et al., 2015*; *Lenoir et al., 2018*) (*Figure 3E*). Further, to account for the possibility that paralogs may not be expressed, we only analyzed genes that were expressed as mRNA in the human cell-line (*Lenoir et al., 2018*) (*Figure 2E*). Using this framework, 35% of conserved human genes contain a fly-to-human paralog, and 407 fly fitness genes (at 5% FDR) are the orthologs of human genes containing at least one paralog. In the human cell-line CRISPR screen, we noted a significant bias against detecting genes that are conserved in flies and have a paralog in the human genome, and this bias was dramatically elevated for the human orthologs of fly fitness genes (*Figure 3F*).

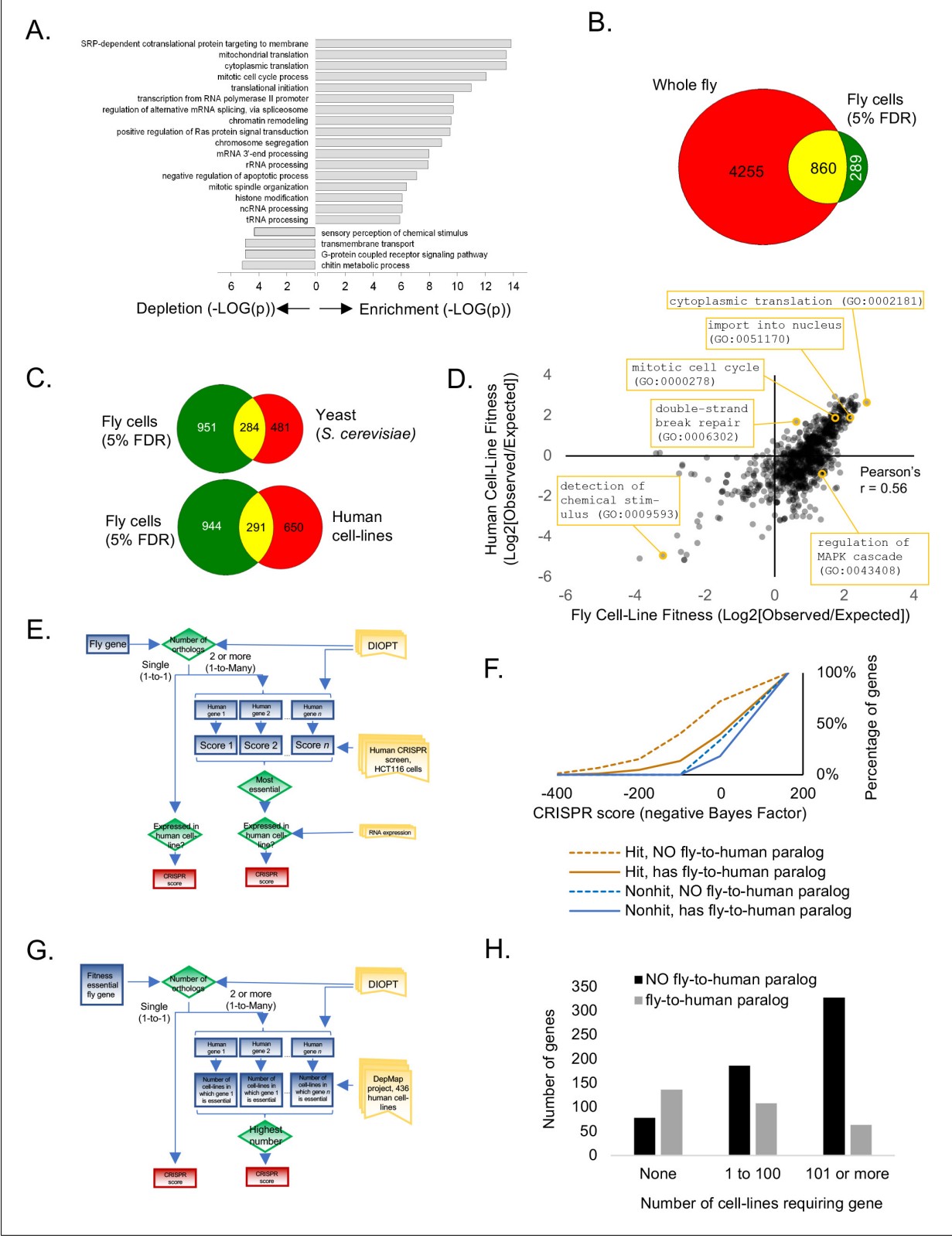

**Figure 3.** Analysis of *Drosophila* S2R+ fitness genes. (**A**) Top enriched gene ontology (GO) terms for screen hits at 5% FDR compared with top enriched GO terms for non-hits. (**B**) Overlap between cell-line CRISPR hits at 5% FDR and all 'lethal' Flybase entries after subtracting entries with no allele information. (**C**) Overlap between *Drosophila* CRISPR hits at 5% FDR and orthologs in yeast (*S. cerevisiae*) or human cell-lines. (**D**) Gene ontology terms enriched in human CRISPR fitness screens (***Hart et al., 2015***) compared with fly CRISPR fitness screens. Selective listing of co-enriched terms, a co-

*Figure 3 continued on next page*

*Figure 3 continued*

depleted term, and outliers. (**E**) Schematic for 'fly-to-human paralog' assignment and testing using high-resolution human CRISPR screen data (*Hart et al., 2015*). For each fly gene with a unique human ortholog, no selection was performed. For fly genes with multiple human orthologs, the most essential human ortholog was chosen. Genes were included in the analysis only if expressed in the human cell-line. Ortholog assignment used DIOPT 'high' and 'moderate' confidence mapping calls (*Hu et al., 2011*). (**F**) Effect of fly-to-human paralogs on hit-calling in human CRISPR screens. Cumulative average of gene fitness essentiality (negative Bayes Factor) for high-resolution human cell-line CRISPR screen (*Lenoir et al., 2018*; *Hart et al., 2015*) examining indicated genesets: those with paralogs are dashed; orthologs of fly fitness genes are brown; orthologs of non-hits are blue. (**G**) Schematic for 'fly-to-human paralog' assignment and testing using cancer Dependency Map data (*Tsherniak et al., 2017*). A CERES score of <0.8 was used for fitness calls. (**H**) Effect of fly-to-human paralogs on number of cell-lines requiring a particular gene for fitness.
DOI: https://doi.org/10.7554/eLife.36333.010

Furthermore, this paralog effect extended to a larger dataset generated by the cancer Dependency Map project, in which CRISPR screens were conducted in 436 human cell-lines. By defining a fitness gene as any gene with a CERES score less than 0.8, we sorted human orthologs of fly fitness by the number of cell-lines out of 436 in which they displayed a fitness defect and according to their paralog relationship between flies and humans (*Figure 3G*). The result again showed a bias against detecting genes with fly-to-human paralogs in the panel of CRISPR screens (*Figure 3H*). Thus, we propose that functional redundancy among closely related genes buffers each of them and makes them invisible in viability screens, and that *Drosophila* cells may be more appropriate for screening genes that expanded during the vertebrate lineage.

Although the *Drosophila* fitness genes we identified are enriched for characterized phenotypes and publication count relative to all genes (data not shown), phenotypes have yet to be described for 303 of them. Among these 303 genes, 251 have human orthologs (*Supplementary file 2*). Thus, further studies of these conserved genes are likely to provide new insights into conserved, cell essential processes not yet studied in flies. Also of interest are fly-specific fitness genes, as they present a paradox and may reveal novel species-specific biology or overlooked structural/functional analogs without sequence orthology and may have potential as targets for new insecticides. At 5% FDR, we obtained 62 fitness genes with no sequence similarity outside of flies using DIOPT, and phenotype information exists regarding 25 of these (*Supplementary file 2*). In confirmation of our methods, these included known cell-essential divergent genes such as *ver* and *HipHop*, which encode components of the telomerin complex, the putative functional analog of mammalian telomerase (*Raffa et al., 2011*; *2010*), and *Kmn1*, *Kmn2* and *Spc105R*, whose gene products may be structural anologs of Ndc80, and Mis12 complex components that interact with centromeric DNA, a proposed driver of speciation (*Schittenhelm et al., 2007*; *2009*; *Henikoff et al., 2001*), as well as several chromatin-interacting proteins (*Supplementary file 2*). Characterization of the remaining conserved and non-conserved genes is likely to bring new insights into essential gene functions.

## Use of *Drosophila* CRISPR screens to uncover gene-drug interactions in the context of major signaling pathway suppression

We next asked whether our CRISPR screening platform could be used to identify genes acting in signaling pathways that regulate cell growth and proliferation (*Friedman and Perrimon, 2007*). To do this, we performed positive selection screens in the presence of trametinib (tra), an inhibitor of the Ras/ERK/ETS pathway, or rapamycin (RAP), an inhibitor of the PI3K/mTor pathway, with the aim of uncovering known and novel compensatory mechanisms or synergizing loss-of-function mutations. Both pro-survival pathways have been extensively mapped through loss-of-function studies in fly tissues and cell-lines (*Friedman and Perrimon, 2007*) (*Figure 4A*). For these experiments, we first transfected cells with Group 1 and Group 2 sublibraries targeting a total of 3974 genes (*Figure 2A*). The gene set interrogated comprises kinases, phosphatases, the fly ortholog of FDA-approved drugs (*Housden et al., 2017*), and fly-to-human paralogs (*Figure 2A*). The cell pools were passaged for 15 days to allow sgRNA integration, subjected to passaging for an additional 30 in sub-lethal doses of tra or RAP, and then re-sequenced (*Figure 2B*). The effect of each drug was carefully monitored by periodically counting cells during the screen to confirm the effect on cell doubling rate (*Figure 4C*).

We observed highly context-specific modes of resistance to each drug. As an illustration, sgRNAs against *aop*, a transcriptional repressor of the Ras/ERK/ETS pathway (*Lai and Rubin, 1992*), or the putative intracellular co-factor for rapamycin, *FK506-bp2* (*Thomson and Johnson, 2010*), conferred

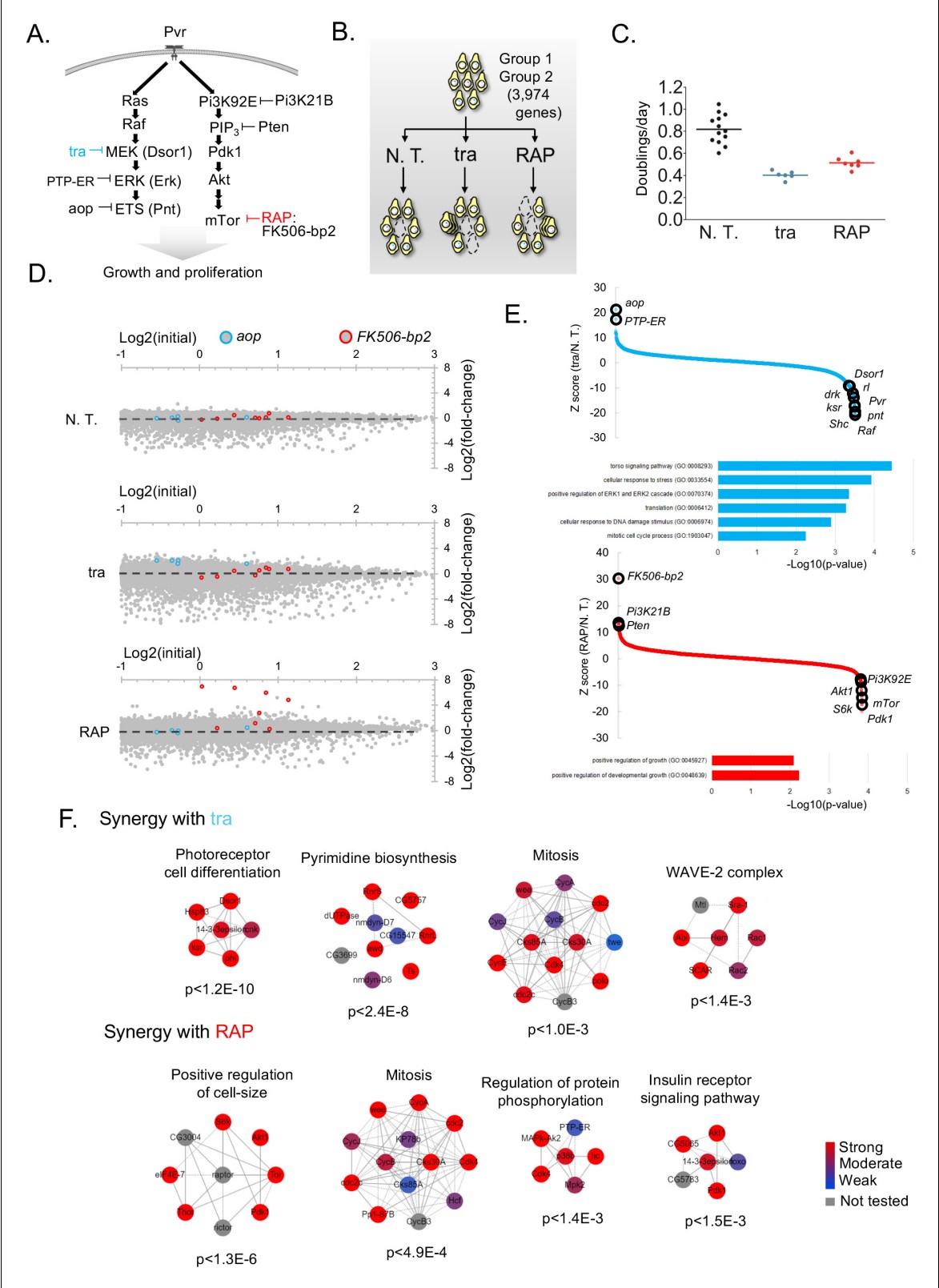

**Figure 4.** Screens to identify genes regulating cell growth and proliferation. (**A**) Schematic of selected components of the Ras/ERK/ETS and PI3K/mTor pathways and of inhibition by trametinib ('tra') or rapamycin ('RAP'). (**B**) Experimental schematic: pathway-specific perturbations to identify context-specific gene essentiality using *Drosophila* CRISPR screens. Dropout screens conducted with no additional treatment (N.T.) serves as a control. (**C**) Estimates of doubling per day obtained during periodic counting of cell pools to verify that tra and RAP partially inhibit cell growth. Each observation

*Figure 4 continued on next page*

*Figure 4 continued*

and mean doubling time plotted. (D) Plot of log2(initial distribution) versus log2(fold-change) for indicated sgRNAs in each screen. Pathway-specific resistance for sgRNAs targeting *aop*, a known suppressor of the Ras/ERK/ETS pathway in *Drosophila* (*Lai and Rubin, 1992*) or *FK506-bp2*, the putative cellular co-factor for rapamycin (*Thomson and Johnson, 2010*). (E) Computed maximum likelihood estimate (MLE) Z score based on sgRNA fold-change data comparing drug treatment condition with no treatment control. sgRNA fold-changes are mean of two independent replicates. Expected intra-pathway negative or positive regulators are noted (see *Supplementary file 3* for complete hit list and raw data). GO terms for synergistic interactions are listed along with hypergeometric p-values for term assignment (PatherDB). (F) Physical protein-protein interaction (PPI) networks enriched using differential CRISPR screens in tra or RAP. PPI network prediction and reported p-values use COMPLEAT, and requires complexes to have >6 members per complex (*Vinayagam et al., 2013*).
DOI: https://doi.org/10.7554/eLife.36333.011

resistance specifically in the context of tra or RAP, respectively (*Figure 4D*). We focused on other such genes for which multiple sgRNAs provide a survival benefit or synergistic lethality in the context of drug treatment using maximum likelihood estimate (*Li et al., 2014*) (MLE) (*Figure 4E*; *Supplementary file 3*). Additional established context-specific pathway negative regulators emerged: PTP-ER, which inactivates Erk (rl) (*Karim and Rubin, 1999*) and Pi3K21B and Pten, which antagonize the activity of Pi3K92E (*Weinkove et al., 1999*; *Huang et al., 1999*; *Goberdhan et al., 1999*).

The analysis also identified several pathway positive regulators as synthetic lethal (*Figure 4E*, and mapped into pathway diagram, *Figure 4A*), cross-pathway synergy, and several novel candidate pathway regulators (*Supplementary file 3*). Gene ontology showed similar but non-overlapping categories were detected as synergistically fitness-compromising with each drug (*Figure 4E*). Moreover, by using differential CRISPR score to map physical protein-protein interaction (PPI) networks, both distinct and overlapping PPI relationships emerged (*Figure 4F*). For instance, a network involved in photoreceptor cell differentiation scored most strongly as synergistically lethal with tra, possibly because photoreceptor differentiation has been a workhorse phenotype for the characterization of mutants in the Ras/ERK/ETS pathway (*Nagaraj and Banerjee, 2004*). Similarly, the top scoring PPI network for RAP-synergy was one involved in the positive regulation of cell-size, a key morphological consequence of upregulating the PI3K/mTOR pathway (*Fingar and Blenis, 2004*). Interestingly, similar PPI networks involved in mitosis was identified in both screens (*Figure 4F*). Taken together, these results demonstrate that CRISPR screening in *Drosophila* cells can reveal context-specific compensatory mechanisms or synergy relevant to major signaling pathways.

## Discussion

CRISPR screening technologies have illuminated the functions of uncharacterized genes and provided a straightforward, cost-effective pipeline to evaluate gene function in different contexts (*Doench, 2018*). However, the benefits of this approach have been inaccessible for *Drosophila* and other insects because the delivery of highly multiplexed DNA libraries has thus far required lentiviral transduction, a process that fails to produce transformed cells (unpublished). Here, we show that multiplexed DNA delivery by transfection followed by site-specific recombination is an effective alternative strategy. We use this technique to deliver a genome-wide library of sgRNA expression cassettes and perform CRISPR knockout fitness screens in *Drosophila* S2R+ cells, identifying 1235 fitness genes at 5% FDR. Of note, our CRISPR screens were conducted after approximately 45 doublings (basal essentiality) or 30 doublings (context-specific screens), while most mammalian screens have been conducted with fewer than 30 doublings. This could possibly due to efficiency differences of the CRISPR systems encoded in the two systems or because of the high ploidy of S2R+ cells. Since our timing optimization data used only two sgRNAs (*Figure 1F*), we do not know how the set of fitness genes would change in screens conducted with fewer doublings. A practical use for viability screens is examining context-specific growth requirements. The *Drosophila* CRIPSR knockout system identified mutants conferring drug resistance or synergy, and should thus be suitable for many future context-specific fitness experiments examining gene-drug/nutrient or gene-gene interactions.

The introduction of systematic knockdown and knockout approaches have greatly reduced false-positive assertions in human cell-line loss-of-function studies, but equally important is knowing what genes are missed by these approaches and providing alternative screening strategies that can target

them. Of critical importance are paralogous genes with redundant functions, i.e., the presence of one paralog buffers against the loss of another (*Ewen-Campen et al., 2017*). Although relevant to human disease (*Dickerson and Robertson, 2012*), these genes are 'invisible' in singe-gene screens and dramatically reduce search space in gene-by-gene screens (*Ewen-Campen et al., 2017*). When we compared human and fly CRISPR screens, human CRISPR screens were less able to detect the fly orthologs of fitness essential genes when those genes had paralogs relative to the fly genome. This supports the use of parallel *Drosophila* and human screens as one approach to offset genetic redundancy.

In addition to CRISPR knockout, the strategy we report can be used with numerous other high-throughput screening modalities which were previously not possible in *Drosophila*, including CRISPR activation, CRIPSR inhibition, CRISPR base-editing, shRNA knockdown, cDNA overexpression, perturb-SEQ, and combinatorial approaches for multigene suppression/activation (*Doench, 2018*; *Najm et al., 2018*; *Shen et al., 2017*). Finally, the methods and constructs we employ are likely to be directly transferable to the large number of existing cultured cell-lines derived from other insects such as mosquitos, where they could be used to characterize viral propagation mechanisms in the vectors of human pathogens such as Dengue or Zika.

## Materials and methods

### Vectors and cell lines

pBS130, phiC31 integrase under control of the HSP70 promoter, was obtained from Addgene. Transient Cas9 expression used pl018 (*Housden et al., 2015*) containing *Drosophila*-optimized Cas9 under the strong Actin promoter. Cas9 from pl018 followed by the BGH terminator from pcDNA3.1 (Invitrogen) were cloned into the SpeI/NotI site of pMK33 (*Koelle et al., 1991*) to generate pMK33/ Cas9. Human codon-optimized intein-Cas9_S219-3XFLAG (*Davis et al., 2015*), a kind gift of D. Lui, was amplified by PCR and cloned into pMK33 to generate pMK33/intein-Cas9_S219-3XFLAG. pLib6.4 was generated by using standard cloning methods as follows. First, the *Drosophila* U6:2 and Act5C promoter cassette from pl018 (*Housden et al., 2015*) was amplified by PCR using primers containing the 5' attB40 site and inserted into pUC19 using Gibson assembly. Next, GFP-T2A-Puro from pAc-STABLE2 (*González et al., 2011*) was amplified with primers containing the 3' attB40 and introduced by ligation into an engineered site in the resulting vector and verified by Sanger sequencing. For minipool experiments in *Figure 1*, sgRNAs were cloned individually using standard methods into the BpiI/BbsI site of pLib6.4 and verified by Sanger sequencing and then mixed. Sequencing reactions were carried out with an ABI3730xl DNA analyzer at the DNA Resource Core of Dana-Farber/Harvard Cancer Center (funded in part by NCI Cancer Center support grant 2P30CA006516-48). S2R+ derivative PT5 (NPT005) was obtained from the *Drosophila* RNAi Screening Center (*Neumüller et al., 2012*) and grown in Schneider's medium (Thermo Fisher Scientific) containing 10% heat-inactivated FBS. PT5 cells were transfected with pMK33/Cas9 or pMK33/intein-Cas9_S219-3XFLAG and selected over a period of two months in 200 ng/uL Hygromycin B (Calbiochem). The resulting PT5/Cas9 cells were maintained in Hygromycin until CRISPR library transfection. pMK33/Cas9 (accession # EvNO00483429), pMK33/intein-Cas9_S219-3XFLAG (accession # EvNO00483430), and pLib6.4 (accession # EvNO00483431) are available through DF/HCC DNA Resource Core (https://plasmid.med.harvard.edu/).

### Library design and construction

In order to allow focused sublibrary screening as in *Figure 3*, sgRNA library was encoded in three separable sublibraries with common controls (Suppl. *Figure 2A*). For gene-targeted sgRNAs, Group 1 targets kinases and phosphatases as assigned using GLAD (*Hu et al., 2015*) and FDA-approved drug-targets (*Housden et al., 2017*). Group 2 targets fly-to-human paralogs, identified by using 'moderate' or 'high' orthology assignment according to DIOPT (*Hu et al., 2011*), not already present in Group 1. Group 3 targets all other remaining genes. Library synthesis was performed by Custom-Array using published methods (*Gilbert et al., 2014*; *Shalem et al., 2014*). Briefly, CRISPR sgRNAs were encoded within a 110-nt single-stranded DNA oligo containing unique library-specific barcode sequences and flanked by BpiI/BbsI sites. 15-cycles of PCR using Phusion Polymerase (New England Biolabs) were used to amplify each sub-library, and then BpiI (Fermentas) was added to the

amplicons. A non-denaturing 20% TBE gel (Thermo) was used to purify the resulting 23-mer fragment and eluted overnight using the crush-soak method. The concentration of ligatable fragments in each preparation was empirically optimized by test ligations in pLib6.4. Optimized concentrations were used in ligations with BpiI-digested pLib6.4 and then electroporated into Ecloni 10GF' electro-competent cells (Lucigen) at a yield of 20–50 times diversity for each library, generating dense colonies on ~60,150 mm LB-carbenicillin plates, which were grown for 18 hr at 37°C and harvested by scraping into LB medium followed by mixing suspended colonies by extensive vortexing. Glycerol was added to 50% and the libraries were flash frozen in 1 mL aliquots. Each sublibrary was prepared by pooled minipreps and eluted in buffer EB (Qiagen). Libraries are available at DRSC/TRiP Functional Genomics Resources (https://fgr.hms.harvard.edu/crispr-cell-screening-reagents).

## Transfection, pool maintenance, and drug addition

pBS130 and pLib6.4 containing CRISPR sublibraries were mixed and co-transfected 1:1 into PT5/Cas9 cells using Effectene with the manufacturer's recommended protocol (Qiagen). For each transfection, cells were first grown until just confluent on T75 flasks for 2–3 days. Doubling was monitored at this passage and determined to be approximately 1/day. Then cells were removed by forceful tapping and replated at $3 \times 10^6$ per well into 6-well dishes. The number of dishes required reflected the library size and accounted for incomplete transfection efficiency to give at least 1500 cells/sgRNA (for reference, a full-genome screen required all wells of eight 6-well dishes). t = 0 used plasmid readcount values. Transfection efficiency and integration efficiency were monitored periodically after transfection for the first twelve days using flow cytometry (BD LSRII). Flow cytometry was performed at the Harvard Medical School Department of Immunology Flow Core. Each 1.5 wells of cells was transferred to one 10 cm dish and passaged in the presence of 5 ug/mL puromycin and cells were allowed to become confluent over 4–6 days. Next, the pool was contracted by two-fold, and $2 \times 10^7$ cells from each of two plates was combined into a single 10 cm dish. $2 \times 10^7$ cells were passaged every 5 days until downstream processing at the indicated time. These steps during early phases of selection ensured no loss of diversity due to variable transfection efficiency. To determine partially inhibitory dose of trametinib (Selleck) or rapamycin (Tocris), the effect of varying doses of each drug was first measured in a 4 day treatment to PT5/Cas9 cells by Cell Titer Glo assay (Promega), using manufacturer's recommended protocol (not shown). From this data, 50 nM trametinib or 2 nM rapamycin was chosen as partially perturbative concentrations for focused library screens (*Figure 3*). To verify that drug treatment decreased doubling rate during the screen, cell counts were performed periodically using a hemocytometer (*Figure 3B*).

## Library sequencing and CRISPR screen data analysis

Genomic DNA was prepared using the Zymo miniprep kit. Assuming DNA content in S2R + cells was 4N, each cell contains ~0.6 pg of DNA. To maintain diversity, all PCRs were conducted from ~5,000 cells per sgRNA per condition. Library amplification was performed using a two-step procedure (illustrated in *Figure 1—figure supplement 3*). First, in-line barcoded inside primers (PCR1F x PCR1R) were used to amplify the library in 23 cycles such that the resulting amplicon had the following sequence: 1/2Read1-$(N)_n$ANNEALINGSEQUENCE-sgRNA-tracrRNA. Primer sequences conformed to: 5'- CTTTCCCTACACGACGCTCTTCCGATCT$(N)_n$ $(B)_6$ gttttcctcaatacttcGTTCg-3' (where N = any nucleotide; n = variable number between 1–9; and B = barcode nucleotide) and 5'- TTTGTGTTTTTAGAATATAGAATTGCATGCTGggtacctc-3'. Next, common outside primers were used to amplify these amplicons with an additional 11 cycles such that the resulting amplicons had the following sequence: P5-Read1-$(N)_n$-ANNEALINGSEQUENCE-sgRNA-tracrRNA-P7. All sequences are provided in *Supplementary file 1*. Following second amplification, amplicons were gel purified and concentration was determined using Qubit dsDNA HS Assay Kit (Thermo). Amplicons were pooled according to concentration of sgRNAs per unit volume. Pooled barcoded amplicons were subjected to sequencing using the NextSeq500 $1 \times 75$ SE platform (Illumina). Sequencing was performed at the CCIB DNA Core Facility at Massachusetts General Hospital (Cambridge, MA) or the Harvard Medical School Biopolymers Facility (Boston, MA). De-multiplexing of the library was performed using TagDust (*Lassmann et al., 2009*) and all downstream analysis was performed within MAGeCK (*Li et al., 2014*) with the following experiment-specific changes: 1) Readcount files were stripped of sgRNAs below the 10th percentile lowest initial readcounts for each sublibrary before

processing in MAGeCK MLE; 2) we performed 1000 iterations for all Z-score assignments. For drug selection experiments, several context-nonspecific sgRNAs providing survival benefit to cells under normal growth conditions were removed due to wide variation of these sgRNAs upon selection.

## Bioinformatics analysis

For ROC curves, major eukaryotic essential complex components for *Drosophila* were from KEGG (http://www.genome.jp/). For gene ontology enrichment analysis, annotation file for *Drosophila* genes was retrieved from NCBI (ftp://ftp.ncbi.nlm.nih.gov/gene/DATA/gene2go.gz). Hypergeometric analysis was done to calculate the enrichment P value using in-house program written in JAVA.

Ortholog assignment as well as fly-to-human protein similarity score were obtained using DIOPT v 6.0.2 using 'moderate' or 'high' confidence calls (*Hu et al., 2011*). RNAseq analysis in this paper used the webtools DGET (http://www.flyrnai.org/tools/dget/web) or CellExpress (http://www.flyrnai.org/cellexpress), which used RNAseq expression data from modENCODE (*Celniker et al., 2009*). For re-analysis of genome-wide arrayed RNAi viability experiment, we re-examined all dsRNA amplicon targets reported in the earlier screen (*Boutros et al., 2004*) using FlyBase v 6. All amplicons with greater than one unique target were discarded (removing 7818 dsRNAs and retaining 13,488). Z-score was then re-calculated using the original methods (*Boutros et al., 2004*). Gene ontology analysis in *Figures 3D* and *4E* was performed using PantherDB (*Mi et al., 2013*). In *Figure 3D*, terms were restricted to those with greater than or equal to 50 members. Complex enrichment analysis in *Figure 4F* used COMPLEAT (*Vinayagam et al., 2013*). In *Figure 4F,* Z-scores in *Supplementary file 3* were multiplied by −1 and top three non-redundant complexes from COMPLEAT with a minimum number of 6 members are reported.

## Data availability

Readcount files for CRISPR analysis compatible with MAGeCK are provided as *Supplementary file 4–15* and *Supplementary file 3*. pMK33/Cas9, pMK33/intein-Cas9_S219-3XFLAG, and pLib6.4. are available through Harvard PlasmID Database (http://plasmid.med.harvard.edu). The three CRISPR sublibraries used in this study are available through DRSC/TRiP Functional Genomics Resources (https://fgr.hms.harvard.edu/).

## Acknowledgements

We thank SE Mohr and BE Housden, and B Ewen-Campen for useful discussions. This work was supported by the LAM foundation (LAM0122P01-17) and the Department of Defense (W81XWH-16-1-0127). RV received support from Harvard Medical School Department of Genetics NIH Training Grant (5T32GM007748-39). YH is supported in part by NIH grant NIGMS R01 GM067761. NP is a Howard Hughes Medical Institute investigator.

## Additional information

### Funding

| Funder | Grant reference number | Author |
| --- | --- | --- |
| LAM Foundation | LAM0122P01-17 | Norbert Perrimon |
| U.S. Department of Defense | W81XWH-16-1-0127 | Norbert Perrimon |
| National Institute of General Medical Sciences | 5T32GM007748-39 | Raghuvir Viswanatha |
| Howard Hughes Medical Institute | | Norbert Perrimon |
| National Institute of General Medical Sciences | R01GM067761 | Norbert Perrimon |

The funders had no role in study design, data collection and interpretation, or the decision to submit the work for publication.

## Author contributions

Raghuvir Viswanatha, Conceptualization, Data curation, Formal analysis, Validation, Investigation, Methodology, Writing—original draft; Zhongchi Li, Validation, Investigation, Methodology, Writing—review and editing; Yanhui Hu, Data curation, Software, Formal analysis, Methodology, Writing—review and editing; Norbert Perrimon, Conceptualization, Supervision, Funding acquisition, Writing—original draft, Project administration, Writing—review and editing

## Author ORCIDs

Raghuvir Viswanatha (iD) http://orcid.org/0000-0002-9457-6953
Norbert Perrimon (iD) http://orcid.org/0000-0001-7542-472X

## Decision letter and Author response

Decision letter https://doi.org/10.7554/eLife.36333.035
Author response https://doi.org/10.7554/eLife.36333.036

---

# Additional files

## Supplementary files

• Supplementary file 1. Fitness essential gene data. Worksheet 2 is a list of all genome-wide sgRNAs. Worksheet 1 contains computed MAGeCK (*Li et al., 2014*) MLE result for each independent replicate of the negative selection screen and the sgRNA-level average. Worksheet 3 contains primer sequences for cloning oligo pools. Worksheet 4 contains primers for primers for amplifying sgRNAs from cell pools (see protocol illustration in *Figure 1—figure supplement 3*).
DOI: https://doi.org/10.7554/eLife.36333.012

• Supplementary file 2. Uncharacterized ('CG') genes and insect-specific fitness essential genes.
DOI: https://doi.org/10.7554/eLife.36333.013

• Supplementary file 3. Context-specific fitness gene essentiality data. Worksheet 1 contains computed MLE result of CRISPR screen conducted in each drug (tra or RAP) versus control. Worksheet 2 contains raw readcount file used to generate data.
DOI: https://doi.org/10.7554/eLife.36333.014

• Supplementary file 4. List file for group 1 Drosophila sgRNA library. File is compatible with MAGeCK (*Li et al., 2014*).
DOI: https://doi.org/10.7554/eLife.36333.015

• Supplementary file 5. List file for group 2 Drosophila sgRNA library. File is compatible with MAGeCK (*Li et al., 2014*).
DOI: https://doi.org/10.7554/eLife.36333.016

• Supplementary file 6. List file for group 3 Drosophila sgRNA library. File is compatible with MAGeCK (*Li et al., 2014*).
DOI: https://doi.org/10.7554/eLife.36333.017

• Supplementary file 7. Readcount file for group 1, replicate 1 of Drosophila sgRNA library following transfection and outgrowth. Column 1 provides internal ID number compatible with *Supplementary file 1*. Column 2 provides targeted gene ID. Column 3, 'REF', provides readcount file from plasmid pool. Column 4 provides readcount following transfection and outgrowth. File is compatible with MAGeCK (*Li et al., 2014*).
DOI: https://doi.org/10.7554/eLife.36333.018

• Supplementary file 8. Readcount file for group 2, replicate 1 of Drosophila sgRNA library following transfection and outgrowth. Column 1 provides internal ID number compatible with *Supplementary file 1*. Column 2 provides targeted gene ID. Column 3, 'REF', provides readcount file from plasmid pool. Column 4 provides readcount following transfection and outgrowth. File is compatible with MAGeCK (*Li et al., 2014*).
DOI: https://doi.org/10.7554/eLife.36333.019

• Supplementary file 9. Readcount file for group 3, replicate 1 of Drosophila sgRNA library following transfection and outgrowth. Column 1 provides internal ID number compatible with *Supplementary file 1*. Column 2 provides targeted gene ID. Column 3, 'REF', provides readcount

file from plasmid pool. Column 4 provides readcount following transfection and outgrowth. File is compatible with MAGeCK (*Li et al., 2014*).

DOI: https://doi.org/10.7554/eLife.36333.020

• Supplementary file 10. Readcount file for group 1, replicate 2 of Drosophila sgRNA library following transfection and outgrowth. Column 1 provides internal ID number compatible with *Supplementary file 1*. Column 2 provides targeted gene ID. Column 3, 'REF', provides readcount file from plasmid pool. Column 4 provides readcount following transfection and outgrowth. File is compatible with MAGeCK (*Li et al., 2014*).

DOI: https://doi.org/10.7554/eLife.36333.021

• Supplementary file 11. Readcount file for group 2, replicate 1 of Drosophila sgRNA library following transfection and outgrowth. Column 1 provides internal ID number compatible with *Supplementary file 1*. Column 2 provides targeted gene ID. Column 3, 'REF', provides readcount file from plasmid pool. Column 4 provides readcount following transfection and outgrowth. File is compatible with MAGeCK (*Li et al., 2014*).

DOI: https://doi.org/10.7554/eLife.36333.022

• Supplementary file 12. Readcount file for group 3, replicate 1 of Drosophila sgRNA library following transfection and outgrowth. Column 1 provides internal ID number compatible with *Supplementary file 1*. Column 2 provides targeted gene ID. Column 3, 'REF', provides readcount file from plasmid pool. Column 4 provides readcount following transfection and outgrowth. File is compatible with MAGeCK (*Li et al., 2014*).

DOI: https://doi.org/10.7554/eLife.36333.023

• Supplementary file 13. Readcount file for average replicates 1 and 2, group 1. Readcounts were internally normalized to a median value of 10000 prior to computing the average. Column 1 provides internal ID number compatible with *Supplementary file 1*. Column 2 provides targeted gene ID. Column 3, 'REF', provides readcount file from plasmid pool. Column 4 provides readcount following transfection and outgrowth. File is compatible with MAGeCK (*Li et al., 2014*).

DOI: https://doi.org/10.7554/eLife.36333.024

• Supplementary file 14. Readcount file for average replicates 1 and 2, group 2. Readcounts were internally normalized to a median value of 10000 prior to computing the average. Column 1 provides internal ID number compatible with *Supplementary file 1*. Column 2 provides targeted gene ID. Column 3, 'REF', provides readcount file from plasmid pool. Column 4 provides readcount following transfection and outgrowth. File is compatible with MAGeCK (*Li et al., 2014*).

DOI: https://doi.org/10.7554/eLife.36333.025

• Supplementary file 15. Readcount file for average replicates 1 and 2, group 3. Readcounts were internally normalized to a median value of 10000 prior to computing the average. Column 1 provides internal ID number compatible with *Supplementary file 1*. Column 2 provides targeted gene ID. Column 3, 'REF', provides readcount file from plasmid pool. Column 4 provides readcount following transfection and outgrowth. File is compatible with MAGeCK (*Li et al., 2014*).

DOI: https://doi.org/10.7554/eLife.36333.026

• Transparent reporting form

DOI: https://doi.org/10.7554/eLife.36333.027

### Data availability

All data generated or analysed during this study are included in the manuscript and supporting files. Readcount files for CRISPR analysis compatible with MAGeCK are provided as Supplementary Files 4-15 and Supplementary File 3. pMK33/Cas9, pMK33/intein-Cas9_S219-3XFLAG, and pLib6.4. are available through Harvard PlasmID Database. The three CRISPR sublibraries used in this study are available through DRSC/TRiP Functional Genomics Resources (https://fgr.hms.harvard.edu/crispr-cell-screening-reagents). Source data files have been provided for Figures 2 (Figure 2—source data 1) and 4 (Supplementary File 3).

The following datasets were generated:

| Author(s) | Year | Dataset title | Dataset URL | Database, license, and accessibility information |
|---|---|---|---|---|
| Raghuvir Viswanatha, Zhongchi Li, Yanhui Hu, Norbert Perrimon | 2018 | pMK33/Cas9 | https://plasmid.med.harvard.edu/PLASMID/GetCloneDetail.do?cloneid=483429&species | Publicly available at the Harvard PlasmID Database (accession no. EvNO00483429) |
| Raghuvir Viswanatha, Zhongchi Li, Yanhui Hu, Norbert Perrimon | 2018 | pMK33/inteinCas9 | https://plasmid.med.harvard.edu/PLASMID/GetCloneDetail.do?cloneid=483430&species | Publicly available at the Harvard PlasmID Database (accession no. EvNO00483430 ) |
| Raghuvir Viswanatha, Zhongchi Li, Yanhui Hu, Norbert Perrimon | 2018 | pLib6.4 | https://plasmid.med.harvard.edu/PLASMID/GetCloneDetail.do?cloneid=483431&species | Publicly available at the Harvard PlasmID Database (accession no. EvNO00483431) |

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
