## [Decision Letter]

Thank you for submitting your article "Pooled genome-wide CRISPR screening for basal and context-specific fitness gene essentiality in *Drosophila* cells" for consideration by *eLife*. Your article has been reviewed by three peer reviewers, one of whom is a member of our Board of Reviewing Editors, and the evaluation has been overseen by Patricia Wittkopp as the Senior Editor. The following individual involved in review of your submission has agreed to reveal his identity: Roderick Beijersbergen (Reviewer #1).

The reviewers have discussed the reviews with one another and the Reviewing Editor has drafted this decision to help you prepare a revised submission.

All three reviewers are in support of publication and the majority comments are minor that could be addressed by rewriting and modifications of the figures. In general, a more extensive discussion on the comparison of *Drosophila* and human CRISPR screening results could be helpful for a broader audience. One of the reviewers requests a clarification of the system itself and data shown in Figure 1B/C that should be addressed by additional comments, analysis or experiments.

– The comparison with human essential genes is interesting but could be explained and illustrated in more detail. In general, the manuscript would gain from a longer discussion on the definition of "essentiality" or "fitness" in the context of their experimental system to illustrate differences in experimental systems (both yeast/human/*Drosophila* as well as RNAi-screening and CRISPR screening). For example, published screens in human cells span a max. incubation time of 30 doublings, whereas the data discussed in this manuscript refers to 45 to 60 cell doublings and more genes are likely to be found to have growth defects within a longer experimental time-frame. The presentation of the differences between human and *Drosophila* cell essentials could also be extended.

– Avoid using the term "de-enrichment" both in text and figures, depletion is an accepted term for loss of abundance in a pooled population. I think the terms “low-redundancy animal cells" (Abstract) and "lower complexity *Drosophila* cells" (“Analysis of *Drosophila* cell fitness genes”) should be rephrased.

– In the Results second paragraph "was transfected along with a plasmid containing" 'along' should be deleted (The phrasing was confusing.)

– In Figure 1B it seems that a large fraction of the cells becomes both mCherry and GFP positive in the presence of the donor and integrase. Would one not expect to lose mCherry expression upon integration of the gRNA cassette containing GFP? This seems a minor fraction of the GFP positive population. I am also not convinced that during the transient transfection of the pLib6.4 donor plasmid and before the integration, no gene editing occurs. The T7E1 assay (Figure 1C) is known for lack of sensitivity and therefore it would be worthwhile to address this in a different way e.g. by gRNA mixing experiments followed by targeted sequencing in a heterogenous population. The targeted sequencing should also be applied on different time-points to get a better insight in the dynamics, which seems rather slow. This could also provide a more satisfying explanation for the observed difference between stable and transient expression of CAS9.

– In Figure 1C the annotation of the FACS plots is missing.

– Figure 1C – 'stable' versus 'transient' should be better explained in the figure legend.

– Figure 2C should indicate correlation of the replicates (e.g. Pearson correlation coefficient of gene-level fold-changes).

– A statistical quantification on the connection between expression and essentiality scoring would be helpful (Figure 2D).

– Legend to Figure 2D: "Z-scores for computed replicate averages" should be explained better.

– Sequence preference motifs of Cas9/sgRNA complex are strongly driven by the binding energy between sgRNA and Cas9 and that the improved scaffold introduced by Qi et al., 2013 removes that bias in most cases (Figure 2H).

– Figure 3C – if the green set are the fly essential genes, then the red set should read "yeast essentials" (top) or "human essentials" (bottom), and not "fly-to-yeast essentials" (which presumably is the yellow overlap?)

– Figure 3C shows the overlap between Dmel, human and *Drosophila* essential genes. The figure legend is very sparse which data sets are being used for comparison (only Hart 2015 or others?). It would also be helpful for the reader to more extensively highlight differences found between species (e.g. which complexes were found / not found). The question is also whether the comparison will change when more essential genes (e.g. from the DepMap project) are being included.

– Figure 4C/E: presentation of context-dependent essential genes could be extended, e.g. by gene enrichment analysis. It should also be explained in the text in more detail how the 3974 genes were chosen.

– Figure labels are often very small and difficult to read.

- The Materials and methods should contain more details regarding library design and construction so that others in the community will be able to implement the screening setup.

– The authors write, "CRISPR sgRNAs were encoded within a 110-nt single stranded DNA oligo containing unique library-specific barcode sequences and flanked by BpiI/BbsI sites." The common sequence of the oligos (5' and 3' to the sgRNA) should be given, including the barcode sequences.

– The sequences of the oligos used for amplifying the sub-libraries should be given.

– Sequencing results of libraries should be made publicly available to allow re-analysis.

Reviewer #1:

The manuscript by Viswanatha et al. describes a CRISPR screening platform in S2R+ *Drosophila* cells based on site specific integration mediated by phiC31. This platform allows for a pooled screening approach with NGS as a read-out, well established in mammalian cells. They have performed a genome-wide lethality screen in S2R+ cells and identify 1235 essential genes. They find depletion of sgRNAs targeting genes in crucial cellular processes analogous to mammalian cell viability CRISPR screen results. Interestingly, but not unexpected, genes with paralogs in human cells are more prevalent in the S2R+ screens than in human cell screens. Finally, the authors provide a proof-of-principle screen for modulators of pathway inhibition in *Drosophila* cells using a MAPK pathway inhibitor Trametinib (MEK) and a PI3K/mTOR pathway inhibitor Rapamycin, identifying several components of the respective pathways.

The establishment of the technology and proof-of-principle examples are well presented and show that large-scale CRISPR screens can be performed in *Drosophila* cells. The identification of lethal genes is relatively easy compared to drug response modifiers, but the examples shown here also illustrate the ability to do so in the S2R+ cells with different treatments. The platform described in this manuscript seems suitable for several types of large scale screens previously performed in mammalian cells and represents a valuable contribution to the scientific community.

I do have some points that need clarification before publication of this work. In Figure 1B it seems that a large fraction of the cells becomes both mCherry and GFP positive in the presence of the donor and integrase. Would one not expect to lose mCherry expression upon integration of the gRNA cassette containing GFP? This seems a minor fraction of the GFP positive population.

I am also not convinced that during the transient transfection of the pLib6.4 donor plasmid and before the integration, no gene editing occurs. The T7E1 assay (Figure 1C) is known for lack of sensitivity and therefore it would be worthwhile to address this in a different way e.g. by gRNA mixing experiments followed by targeted sequencing in a heterogenous population. The targeted sequencing should also be applied on different time-points to get a better insight in the dynamics, which seems rather slow. This could also provide a more satisfying explanation for the observed difference between stable and transient expression of CAS9.

The authors claim that *Drosophila* cells "do not have detectable copy-number bias" but do not provide an explanation. It could be that the genome editing is inefficient and not resulting in multiple double strand breaks, preventing a lethal DNA damage mediated cellular response. It would be informative to show the increased numbers of DNA damage foci in cells expressing gRNAs targeting copy-number-amplified loci and study the response to double strand breaks in SR2+ cells.

Reviewer #2:

In the submitted manuscript, Viswanatha and colleagues established the methodology and for the first time performed pooled CRISPR-knock out screens in a *Drosophila* cell line. So far, pooled CRISPR-Cas9 in insect cells was not possible because of lack of a suitable methodology for efficient integration of sgRNA plasmids. To circumvent the inefficiency of lentiviral delivery to *Drosophila* cells, the authors established a site-specific library integration strategy and first confirmed its suitability using a limited number of target sgRNAs. They then designed and performed a genome-wide CRISPR knock-out screen in S2R+ cells and identified approx. 1200 essential genes at a 5% FDR. The benchmarked this data set against previous RNAi screens and CRISPR screens in human cells. An important finding is that they could identify a significant number of genes (approx. 400) that are possibly not lethal in human cells because of gene duplication. They further demonstrate that this screening platform is suitable for chemical-genetic screens and show data on MEK and mTOR inhibitor screens using a sub library of 3974 genes.

Overall, this is a well executed and written study. It is the first time that a pooled screening approach has been (successfully) used in insect cells, opening new avenues for genetic screening. The site-specific integration strategy is elegant and circumvents problems with viral or other types of delivery – this is also an important methodological advance that will open new experimental avenues to perform other types of over expression and silencing studies in a pooled format in *Drosophila* cells. The part on the comparison with human essential genes is interesting but could be explained and illustrated in more detail. In general, the manuscript would gain from a longer discussion on the definition of "essentiality" or "fitness" in the context of their experimental system to illustrate differences in experimental systems (both yeast/human/*Drosophila* as well as RNAi-screening and CRISPR screening). For example, published screens in human cells span a max. incubation time of 30 doublings, whereas the data discussed in this manuscript refers to 45 to 60 cell doublings and more genes are likely to be found to have growth defects within a longer experimental time-frame. The presentation of the differences between human and *Drosophila* cell essentials could also be extended.

Reviewer #3:

Viswanatha et al. describe a method for performing pooled genome-wide CRISPR screens in Drosophils S2R+ cells. Using this, they identify 1235 genes as influencing cell fitness in S2R+ cells.

Overall, the data are of good quality. No groundbreaking new biological insights have come out of this manuscript – it is mainly a methods paper, but the method will surely be useful to the community.

I have only a few minor comments.

1. The Materials and methods should contain more details regarding library design and construction so that others in the community will be able to implement the screening setup.

- the authors write "CRISPR sgRNAs were encoded within a 110-nt single stranded DNA oligo containing unique library-specific barcode sequences and flanked by BpiI/BbsI sites." The common sequence of the oligos (5' and 3' to the sgRNA) should be given, including the barcode sequences.

- the sequences of the oligos used for amplifying the sub-libraries should be given

2. In the Results second paragraph "was transfected along with a plasmid containing": 'along' should be deleted (The phrasing was confusing.)

3. Figure 1C - 'stable' versus 'transient' should be better explained in the figure legend.

4. Legend to Figure 2D: "Z-scores for computed replicate averages" should be explained better

5. Figure 3C - if the green set are the fly essential genes, then the red set should read "yeast essentials" (top) or "human essentials" (bottom), and not "fly-to-yeast essentials" (which presumably is the yellow overlap?)

---

## [Author Response]

All three reviewers are in support of publication and the majority comments are minor that could be addressed by rewriting and modifications of the figures. In general, a more extensive discussion on the comparison of Drosophila and human CRISPR screening results could be helpful for a broader audience. One of the reviewers requests a clarification of the system itself and data shown in Figure 1B/C that should be addressed by additional comments, analysis or experiments.– The comparison with human essential genes is interesting but could be explained and illustrated in more detail. In general, the manuscript would gain from a longer discussion on the definition of "essentiality" or "fitness" in the context of their experimental system to illustrate differences in experimental systems (both yeast/human/Drosophila as well as RNAi-screening and CRISPR screening). For example, published screens in human cells span a max. incubation time of 30 doublings, whereas the data discussed in this manuscript refers to 45 to 60 cell doublings and more genes are likely to be found to have growth defects within a longer experimental time-frame. The presentation of the differences between human and Drosophila cell essentials could also be extended.

We now describe the overlap in gene ontology terms in greater detail between fly and human fitness genes (Figure 3D). We have added a section to the Discussion as follows: “Of note, our CRISPR screens were conducted after approximately 45 doublings (basal essentiality) or 30 doublings (context-specific screens), while most mammalian screens have been conducted with fewer than 30 doublings. Since our timing optimization data used only two sgRNAs (Figure 1F), we do not know how the set of fitness genes would change in screens conducted with fewer doublings.”

– Avoid using the term "de-enrichment" both in text and figures, depletion is an accepted term for loss of abundance in a pooled population.

We have made the changes.

I think the terms "low-redundancy animal cells" (Abstract) and "lower complexity Drosophila cells" (“Analysis of Drosophila cell fitness genes”) should be rephrased.

We have removed the terms in both cases.

– In the Results second paragraph "was transfected along with a plasmid containing" 'along' should be deleted (The phrasing was confusing.)

We removed the word ‘along’.

– In Figure 1B it seems that a large fraction of the cells becomes both mCherry and GFP positive in the presence of the donor and integrase. Would one not expect to lose mCherry expression upon integration of the gRNA cassette containing GFP? This seems a minor fraction of the GFP positive population.

Generation of almost all of the GFP-positive population is integrase-dependent because of our analysis comparing the integration efficiency with or without phiC31 integrase. Therefore, we believe there are two *attP* sites present in our *attP* line (PT005). This is supported by 1) a decrease in GFP expression between the median of GFP-RFP double-positives versus GFP single positives, 2) because a single intronic insertion was observed by inverse PCR sequencing of the cell-line, and 3) because CGH analysis by others shows that there are two copies of the X chromosome, which is the chromosome that contains the intron into which the *attP* cassette site inserted.

I am also not convinced that during the transient transfection of the pLib6.4 donor plasmid and before the integration, no gene editing occurs. The T7E1 assay (Figure 1C) is known for lack of sensitivity and therefore it would be worthwhile to address this in a different way e.g. by gRNA mixing experiments followed by targeted sequencing in a heterogenous population. The targeted sequencing should also be applied on different time-points to get a better insight in the dynamics, which seems rather slow. This could also provide a more satisfying explanation for the observed difference between stable and transient expression of CAS9.

We do not believe that no editing occurs initially leading to multiple edits, but rather that the quantitative number of early edits is tolerably low (before unintegrated plasmids are lost) compared to the number later in the experiment, at which point most of the cells have 1 or 2 sgRNAs/cell. Because sgRNAs mix randomly, we suspect that a low level of initial editing will not result in detectable loss of signal-to-noise. We do not believe that there is a practical way to resolve this by sequencing clonal isolates.

– In Figure 1C the annotation of the FACS plots is missing.

We now include annotations.

– Figure 1C – 'stable' versus 'transient' should be better explained in the figure legend.

We now phrase this differently: “Cells stably or transiently transfected to express Cas9 or control vector were each additionally transfected with an sgRNA targeting the Dredd allele followed by editing efficiency assay

(T7E1) at the Dredd locus.”

– Figure 2C should indicate correlation of the replicates (e.g. Pearson correlation coefficient of gene-level fold-changes).

We have now included a Pearson’s correlation coefficient, which is 0.65, indicating a strong correlation between replicates.

– A statistical quantification on the connection between expression and essentiality scoring would be helpful (Figure 2D).

We performed false-discovery rate assessment according based on rank. The formula used is now included within Figure 2E.

– Legend to Figure 2D: "Z-scores for computed replicate averages" should be explained better.

We now change the text to indicate the order of operations: that the average of all replicate log2(foldchanges) was first computed, and then Z-score assessment was performed on this data.

– Sequence preference motifs of Cas9/sgRNA complex are strongly driven by the binding energy between sgRNA and Cas9 and that the improved scaffold introduced by Qi et al., 2013 removes that bias in most cases (Figure 2H).

We cannot locate the reference indicated by the reviewer, as Qi et al., 2013 does not contain improvements to the scaffold. We located a paper in which Qi is a middle author, Chen et al., 2013, which does discuss alterations to the guide scaffold that improve CRISPRi, but we cannot locate a reference that suggests that it removes sequence motif preferences in the context of CRISPR KO screens. As we note in Figure 2H, our position matrix analysis for high performing sgRNAs identified 5/9 positions in common with human CRISPR screens analyzed less than two years ago (Doench et al., 2016). The high degree of overlap between the methods suggests that our screens have similar biases compared with up-to-date mammalian CRISPR screening methods. It would be an interesting point to compare the altered guide scaffold, and we are considering potential improvements along these lines for future studies.

– Figure 3C – if the green set are the fly essential genes, then the red set should read "yeast essentials" (top) or "human essentials" (bottom), and not "fly-to-yeast essentials" (which presumably is the yellow overlap?)

Accepted and revised.

– Figure 3C shows the overlap between Dmel, human and Drosophila essential genes. The figure legend is very sparse which data sets are being used for comparison (only Hart 2015 or others?). It would also be helpful for the reader to more extensively highlight differences found between species (e.g. which complexes were found / not found). The question is also whether the comparison will change when more essential genes (e.g. from the DepMap project) are being included.

Regarding major complexes or biological processes, our analyses suggest that they are largely unaltered between fly and human essential genes, even though the identities of the genes themselves are different. In the previous version of the manuscript, we did not show data to support this. We now include the enrichment of GO terms using Pather DB, which allows us to directly compare terms between fly and human gene sets (Figure 3D), and find high overlap between the terms (Pearson’s r = 0.56). We note that for the analysis, we only include GO terms that have greater than 50 members as this excludes statistical noise due to small gene sets, leading to stochastic variation. With all GO terms, the Pearson’s correlation is still >0.50, suggesting that there is a strikingly robust relationship between GO terms enriched from fitness screens in human and fly cell-lines.

We thank the reviewer for the suggestion to compare DepMap human cell-line CRISPR screen data, which was released in raw form just last month. This more extensive comparison illustrates the paralog redundancy effect for the human orthologs of our *Drosophila* cell-line essential genes using a larger panel of 436 human cell-lines (Figure 3F). The conclusion of this analysis is largely the same.

– Figure 4C/E: presentation of context-dependent essential genes could be extended, e.g. by gene enrichment analysis.

Very few genes enrich following treatment (positive-selection), and the high confidence hits are all listed in the figure. However, genes that are synergistic with the treatment were listed selectively, so we agree that a more extensive analysis is worthwhile. We now address this by 1) performing GO enrichment for the synergistic candidates (Figure 4E), 2) protein complex enrichment for the candidates using COMPLEAT (flyrnia.org/COMPLEAT) (Figure 4F). The most striking conclusions were from protein complex enrichment: genes linked to photoreceptor cell proliferation (where many Ras pathway components were first identified genetically in flies) were top hits for synergistic lethality with trametinib, whereas genes linked to positive regulation of cell-size (a well-studied consequence of Tor activity) were the top hit for rapamycin synergy. We believe that these analyses greatly strengthen the conclusion that *Drosophila* CRISPR screens can be used to identify genes whose perturbation may function synergistically with drugs targeting cell proliferation pathways.

It should also be explained in the text in more detail how the 3974 genes were chosen.

The genes used in the context-dependent screens were from the group1 and group2 sublibraries. We have now included more details in the Materials and methods section regarding gene selection. We write: “For gene-targeted sgRNAs, Group 1 targets kinases and phosphatases as assigned using GLAD (Hu et al., 2015) and FDA-approved drug-targets (Housden et al., 2017). Group 2 targets fly-to-human paralogs, identified by using “moderate” or “high” orthology assignment according to DIOPT (Hu et al., 2011), not already present in Group 1. Group 3 targets all other remaining genes.” We also include a statement about Group1 and Group2 gene selection in the Results section.

– Figure labels are often very small and difficult to read.

We have increased the font sizes where possible.

- The Materials and methods should contain more details regarding library design and construction so that others in the community will be able to implement the screening setup.

We now include all primer sequences in Supplementary File 1, and we added a new Figure 1—figure supplement 3, which diagrams the library vector and sequencing procedure and should make it possible for others in the community to perform these experiments.

– The authors write, "CRISPR sgRNAs were encoded within a 110-nt single stranded DNA oligo containing unique library-specific barcode sequences and flanked by BpiI/BbsI sites." The common sequence of the oligos (5' and 3' to the sgRNA) should be given, including the barcode sequences.– The sequences of the oligos used for amplifying the sub-libraries should be given.

We now include all primer sequences in Supplementary File 1 including those used to prepare the library and sequence the integrated library from cell pools, and we have added a figure (Figure 1—figure supplement 3) which includes a map of the library vector and outlines the sequencing procedure.

– Sequencing results of libraries should be made publicly available to allow re-analysis.

We will coordinate with *eLife* to make these files available.

Reviewer #1:The authors claim that Drosophila cells "do not have detectable copy-number bias" but do not provide an explanation. It could be that the genome editing is inefficient and not resulting in multiple double strand breaks preventing a lethal DNA damage mediated cellular response. It would be informative to show the increased numbers of DNA damage foci in cells expressing gRNAs targeting copy-number-amplified loci and study the response to double strand breaks in SR2+ cells.

We agree with the reviewer that the kinetics of creating the double-strand breaks in our system may limit the DNA damage response and therefore the effect of copy number bias in these screens, and this may be a mechanism for why we cannot detect the bias. However, we do not believe monitoring the response to DNA damage is within the scope of the aims of the study. Interestingly, the reviewer’s hypothesis that human cell line CRIPSR is more sensitive to double-strand breaks is supported by a new analysis we present in Figure 3D. Here, we show that while most GO terms between fly and human CRISPR fitness screens are similar, one outlier is “double-strand break repair (GO:0006302),” which is captured by the human fitness genes to a greater extent than the fly fitness genes. This may hint that human cell-lines treated with CRISPR-induced breaks are more sensitive to the loss of DNA repair machinery than the *Drosophila* cell-line, and this may have to do with overall cutting efficiency or kinetic differences in the two systems.

Reviewer #2:Overall, this is a well executed and written study. It is the first time that a pooled screening approach has been (successfully) used in insect cells, opening new avenues for genetic screening. The site-specific integration strategy is elegant and circumvents problems with viral or other types of delivery – this is also an important methodological advance that will open new experimental avenues to perform other types of over expression and silencing studies in a pooled format in Drosophila cells. The part on the comparison with human essential genes is interesting but could be explained and illustrated in more detail. In general, the manuscript would gain from a longer discussion on the definition of "essentiality" or "fitness" in the context of their experimental system to illustrate differences in experimental systems (both yeast/human/Drosophila as well as RNAi-screening and CRISPR screening). For example, published screens in human cells span a max. incubation time of 30 doublings, whereas the data discussed in this manuscript refers to 45 to 60 cell doublings and more genes are likely to be found to have growth defects within a longer experimental time-frame. The presentation of the differences between human and Drosophila cell essentials could also be extended.

We thank the reviewer for bringing up this important point. We now include a paragraph about essentiality versus fitness in the Introduction.